# Research on Height Adjustment Characteristics of Heavy Vehicle Active Air Suspension Based on Fuzzy Control

Xin Bai, Liqun Lu *, Can Zhang and Wenpeng Geng

School of Transportation and Vehicle Engineering, Shandong University of Technology, Zibo 255000, China; 21402030146@stumail.sdut.edu.cn (X.B.); 17120202039@stumail.sdut.edu.cn (C.Z.); 21503030304@stumail.sdut.edu.cn (W.G.)
* Correspondence: luliqun@sdut.edu.cn

**Abstract:** The suspension system's performance has a direct impact on the ride comfort, handling stability, and driving safety of heavy vehicles. The active air suspension of heavy vehicles can adjust the stiffness, damping parameters, and body height in real-time based on different road conditions. This adjustment ensures that the entire vehicle experiences a smooth ride while also making vehicle loading and unloading more labor efficient. Additionally, the active air suspension system enables the vehicle to achieve automation and intelligence. This study focused on a particular $6 \times 4$ heavy vehicle and designed an active air suspension system that aligns with the vehicle parameters of the heavy truck. Through the use of a fuzzy PID active control strategy, this study investigated and analyzed the height adjustment of the air spring. The results indicate that at a vehicle speed of 60 km/h on a class A road surface, the vehicle body's vertical acceleration was reduced by 22.1%, and the dynamic travel of the suspension was reduced by 20.1%. This indicates that the fuzzy PID active air suspension system effectively reduces the vehicle's vibrations and improves ride comfort.

**Keywords:** heavy vehicle; active air suspension; fuzzy PID; control strategy; simulation

## 1. Introduction

The suspension system is a crucial component of a heavy vehicle's chassis, as it greatly impacts the ride comfort. An active air suspension system adjusts its stiffness and damping in real time, based on the uneven and bumpy road conditions, enabling the vehicle to achieve optimal driving conditions. The active air suspension system possesses excellent shock absorption performance and stability, significantly reducing the cab's heaving and the trailer's shaking during heavy vehicle driving, minimizing cargo damage and reducing driver fatigue. The use of air suspension in heavy trucks, especially heavy truck traction vehicles, is widespread in countries with developed automotive industries such as Europe, America, and Japan. It is predicted that in the next few years, air suspension systems will become the mainstream configuration of heavy vehicles [1–4].

Researchers from various countries have conducted extensive studies on active air suspension systems, including control strategies and other related aspects. For instance, Christian Graf et al. designed a force-controlled air spring for active suspension in commercial vehicle cockpits, whereby the damping forces of the air spring are generated by varying the air mass within its volume [5]. Nguyen Van Tuan et al. aimed to improve the performance of the semi-trailer air suspension system by taking the dynamic load coefficient (DLC) as the objective function, discussing the optimization design method of the geometric parameters of the air spring suspension system [6]. Wang Shaohua et al. from Jiangsu University completed the vehicle modeling of the air suspension of a bus and established a complete multibody dynamic model of the vehicle. They analyzed the matching damping of the air spring suspension, selecting body vertical acceleration, suspension working space, and dynamic tire load as performance indices in four typical working conditions [7]. Cheng Yue, a doctoral student at Jilin University, conducted an

optimization and experimental study on the parameter matching of the bus air suspension system, and the results proved the correctness and practicality of the optimal control method [8–13]. Yan Tianyi et al. conducted comprehensive testing on the air suspension system, including stiffness testing, damping testing, and on-road vehicle testing. Through in-depth analysis and experimentation on the stiffness characteristics of the air suspension system, they fitted the relationship curve between the stiffness of the air spring and the inflation/deflation time. In the damping testing, they compared and evaluated different testing methods, finding that using the basic excitation method yielded more realistic damping characteristics of the air spring. The real vehicle testing validated the authenticity and effectiveness of the control strategy, providing valuable insights for the control strategy of the electronic-controlled air suspension system [14–17]. Li Hailin et al. conducted an analysis and research on the air suspension of heavy-duty tractor trucks using the multibody dynamics software ADAMS. They optimized the layout and basic parameters of the guiding rod system in the air suspension and further performed vehicle optimization using ADAMS/Insight, resulting in a more rational arrangement of the air suspension guiding rod system. Additionally, they conducted theoretical analysis and utilized the orthogonal experimental method to study the design and performance of the air springs, providing solutions for improving aspects such as the vehicle's brake nodding, acceleration squat, and lateral stability [18–23]. To evaluate the performance of control damping (CD) and control air spring (CAS) of the vehicle air suspension system regarding driving comfort and road friendliness, Nguyen V. et al. proposed a three-dimensional (3D), nonlinear dynamic model of heavy trucks with 14 degrees of freedom (DOF) and an optimal fuzzy control (OFC) with control rules optimized by genetic algorithms (GAs) [24]. Yang Chen et al. designed the traditional air suspension to improve the dynamic response of the suspension body to body roll, enhancing the vehicle's handling and stability [25]. Lastly, Ma Xinbo et al. considered the vehicle dynamics as a highly nonlinear model, proposing a new nonlinear predictive controller to deal with the multi-objective control requirements of the vehicle system [26]. Tuan Anh Nguyen et al. conducted research on methods for controlling active suspension system operation, namely the AFSPIDF algorithm, OSMC algorithm, and integrated pneumatic suspension system. The AFSPIDF algorithm combines PID, SMC, and Fuzzy algorithms, significantly reducing vehicle vibration displacement and acceleration, thereby enhancing vehicle stability and comfort. The OSMC algorithm controls vehicle vibration using a quarter-dynamic model, maintaining stable interaction between the wheels and the road surface. The integrated pneumatic suspension system integrates pneumatic suspension and hydraulic actuators, resulting in a significant reduction in displacement and acceleration of the sprung mass under random excitation. These research findings demonstrate the advantages of these methods over traditional approaches, substantially improving vehicle performance and comfort [27–29]. These studies mainly focused on the structural design, control strategies, modeling, and simulation of air suspension or active suspension, with less involvement in the control system design and control strategy research for active air suspension in heavy-duty vehicles.

This paper presents the design of an active air suspension system for a 6 × 4 heavy vehicle, along with a detailed study and analysis of the height regulation control strategy of the active air suspension. It not only addresses the key issue of customizing an active air suspension system for heavy vehicles but also introduces an advanced fuzzy PID control strategy, setting a new standard for enhancing ride comfort and overall driving performance. The innovative combination of fuzzy logic and PID control bestows adaptability and intelligence upon the suspension system, paving the way for the continuous evolution of heavy vehicle technology towards a safer, more efficient, and comfortable future. The structure of this paper will unfold as follows. Firstly, Section 1 provided an introduction, introducing the background and objectives of this study, emphasizing the significance and necessity of active air suspension systems in the field of automotive engineering. Next, Section 2 will comprehensively describe the design of the active air suspension system, including its working principles, structural design, and the design of the air suspension

height adjustment controller. In Section 3, we will elucidate the modeling process of the active air suspension system, encompassing the establishment of the pavement excitation model and the air suspension model, along with an introduction to the design method of the fuzzy PID controller. Subsequently, Section 4 will present the simulation results and in-depth analysis, evaluating the performance of the active air suspension system under various operating conditions. Lastly, Section 5 will summarize the research findings, provide conclusions and discussions, highlighting the research contributions of this paper and potential directions for future expansion. Through the aforementioned chapter arrangement, we aim to systematically present the research content and outcomes of this study, offering readers a clear guide to the paper's structure.

## 2. Design of Active Air Suspension System

### 2.1. Working Principle of Active Air Suspension System

An active suspension is a suspension system that actively adjusts the suspension stiffness and damping based on the vehicle's state and road conditions. It uses sensors to sense the vehicle's motion and road information and adjusts the suspension characteristics in real time through a controller. On the other hand, an air suspension utilizes an air pressure adjustment system to change the suspension height and stiffness by increasing or decreasing the gas pressure in the air springs. An active air suspension is an advanced suspension system that integrates both active suspension and air suspension technologies. It combines the advantages of active suspension and air suspension to provide advanced suspension performance and driving comfort. As shown in Figure 1, the active air suspension is used on the drive axle and mainly includes the following parts: sensor and control unit, power source, and executive elements (air spring and shock absorber).

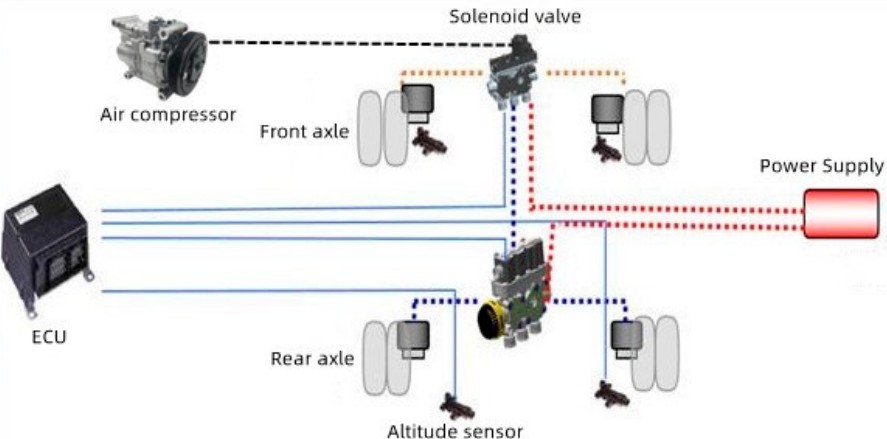

**Figure 1.** Composition of active air suspension system.

The active air suspension system comprises an electronic control unit (ECU) as its central component. The key sensors involved in the system are the body height sensor and body acceleration sensor. The ECU processes signals from these sensors to regulate the air pump's operation, generating active control forces through fuzzy PID processing. This approach helps to optimize the control signal, leading to the best possible performance of the system.

The electronic control unit (ECU) regulates the solenoid valve's opening and closing, adjusting the current to transmit signals to the actuating element. This helps to fine-tune the air spring pressure and shock absorber damping force, optimizing the suspension stiffness and damping. Since the air spring medium is air, the air source requires careful consideration. The ECU controls the air compressor to compress the air and provide compressed air for the airbag. The air compressor is linked to the airbag by pipes, with various solenoid valves distributed in between to facilitate the system's operation.

### 2.2. Structural Design of Active Air Spring System

In our research, we have chosen a specific type of 6 × 4 heavy-duty vehicle with vehicle parameters as shown in Table 1 and Figure 2.

**Table 1.** Vehicle parameters of heavy-duty vehicle.

| Name | Numerical Value | Name | Numerical Value |
|---|---|---|---|
| Wheelbase | 3200 + 1400 mm | Total mass (full load) | 25 t |
| Allowable load of front axle | 7000 kg | Vehicle weight (No load) | 8.8 t |
| Permissible rear axle load | 18,000 kg | Body length | 6985 mm |
| Front track width | 2041 mm | Body width | 2496 mm |
| Rear track width | 1830 mm | Body height | 3850 mm |

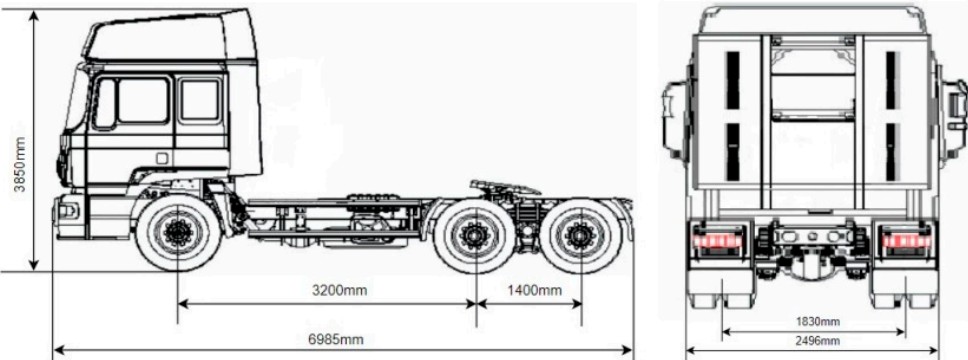

**Figure 2.** Vehicle dimensions of heavy-duty vehicle.

This particular 6 × 4 heavy vehicle utilizes leaf springs for its front axle, while the middle and rear axles use air springs. Each single axle is equipped with four air springs. The basic parameters of the rear axle suspension can be found in Table 2:

**Table 2.** Basic parameters of double-axle eight-airbag suspension.

| Name | Numerical Value |
|---|---|
| Design load (full load) | 18,000 kg |
| Suspension height | 210 mm |
| Double-axle wheelbase | 1400 mm |
| Suspension width | 762 mm |
| Drive axle elevation | 3° |

To determine the appropriate parameters for the active air suspension, the vehicle parameters of a 6 × 4 heavy vehicle and the basic parameters of a double-axle eight-airbag suspension were taken into account. An active air suspension system typically consists of various components, such as a sensor and control unit, power source, actuator (air spring and shock absorber), and a force and torque generating actuator. Figure 3 shows the three-dimensional structure diagram of the double-axle eight-airbag active air suspension system for heavy vehicles.

The upper member of the whole air suspension is a V-shaped thrust rod, and the lower member is a U-shaped stabilizer rod, that is, an upper "V" and a lower "U" structure. The two air springs on the same side of a single axle are connected by brackets. The design structure of the rear axle suspension in the double-axle eight-airbag system has good structural symmetry. For the convenience of research, the single-axle four-airbag air suspension was taken as the research object.

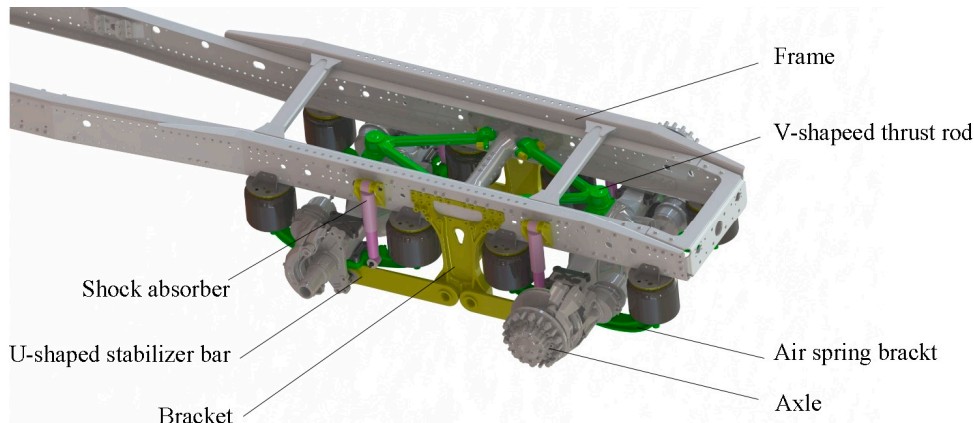

**Figure 3.** Double-axle eight-airbag air suspension system.

### 2.3. Design of Air Suspension Height Adjustment Controller

After analyzing the air spring's basic characteristics and inflation/deflation features, we can conclude that air springs are characterized by variable stiffness, which is lower than that of leaf springs. By adjusting their inflation and deflation levels, air springs can alter their stiffness and adjust the body height of heavy vehicles accordingly. Based on these characteristics, we designed a closed-loop control diagram for the double-axle eight-airbag active air suspension system's body height, as shown in Figure 4. The control method employed is the three "points" control approach. This means that the middle axle's four air springs (air springs 5, 6, 7, and 8) share a height sensor, while the rear axle's two air springs (air springs 1 and 2, air springs 3 and 4) on the same side share a height sensor. Each air spring is connected to a solenoid valve, which the ECU uses to regulate the air spring's charging and discharging by controlling the solenoid valve's opening. This allows the system to achieve intelligent height control under different road conditions.

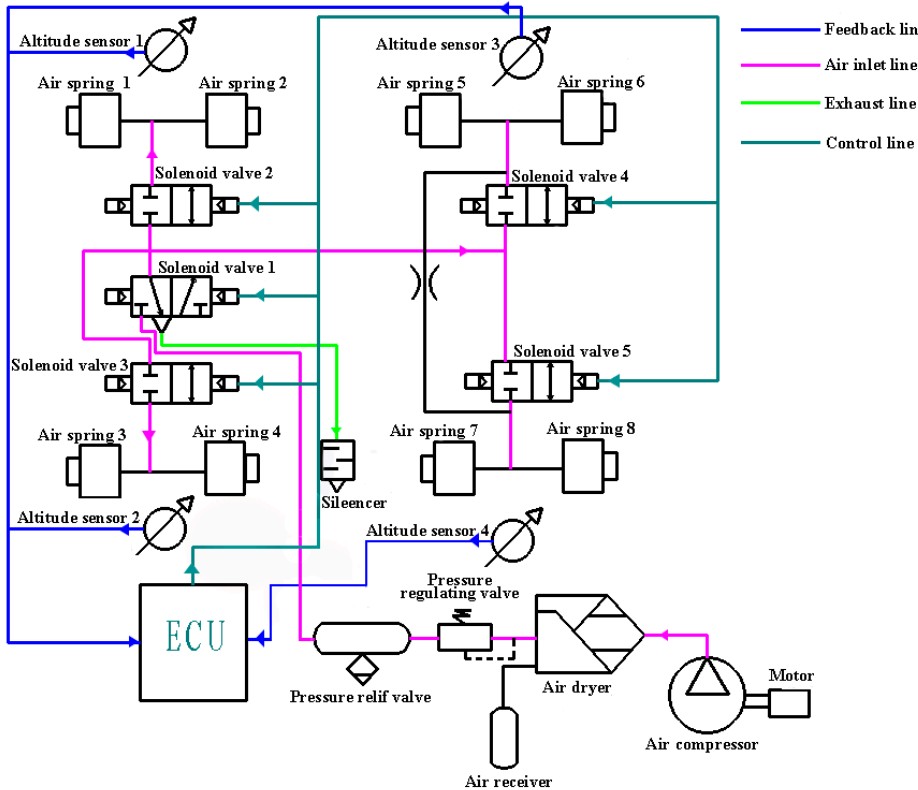

**Figure 4.** Schematic diagram of body height control of double-axle eight-airbag active air suspension.

## 3. Active Air Suspension System Modeling

### 3.1. Establishment of Pavement Excitation Model

3.1.1. Random Signal Road Excitation

Random signal pavement excitation can truly reflect the pavement conditions, and the statistical characteristics are generally described by the pavement power spectral density. According to ISO [30], Formula (1) is the fitting expression of the power spectral density.

$$G_q(n) = G_q(n_0)\left(\frac{n}{n_0}\right)^{-w} \tag{1}$$

where $n$—spatial frequency, $n_0$—reference spatial frequency, and $G_q(n_0)$—pavement roughness coefficient, which is the pavement power spectral density in the $n_0$ state, $\sigma_q$—standard deviation of road roughness, and $w$—frequency index.

The surface roughness can be divided into eight grades A–H. When $n_0 = 0.1$ m$^{-1}$ and $w = 2$ in Formula (1), the geometric average value of pavement roughness of each pavement grade is shown in Table 3 [31].

**Table 3.** Pavement roughness grade.

| Different Pavement Grades | $G_q(n_0)/(10^{-6}\text{m}^3)$ | $\sigma_q/(10^{-6}\text{m}^3)$ |
|:---:|:---:|:---:|
| | Geometric Mean | Geometric Mean |
| A | 16 | 3.81 |
| B | 64 | 7.61 |
| C | 256 | 51.23 |
| D | 1024 | 30.45 |
| E | 4096 | 60.90 |
| F | 16,384 | 121.80 |
| G | 65,336 | 243.61 |
| H | 262,144 | 487.22 |

The driving speed of heavy-duty vehicles is also a key factor in the dynamic characteristics of the active air suspension system, and the spatial frequency power spectral density $G_q(n)$ cannot reflect the key factor of vehicle speed well. Therefore, the concept of time frequency is introduced. The relationship between the time frequency f and the spatial frequency n is as follows:

$$f = vn \tag{2}$$

where $v$—vehicle speed, unit: m/s.

According to Formula (2), the expression of the time–frequency power spectral density $G_q(f)$ is as follows:

$$G_q(f) = \frac{G_q(n)}{v} \tag{3}$$

When $w$ is a fixed value of 2, the expression of the time–frequency power spectral density can be written as:

$$G_q(f) = G_q(n_0){n_0}^2\frac{v}{f^2} \tag{4}$$

In combination with engineering practice, when the spatial frequency $n$ is generally taken as 0.011 m$^{-1}$, it is more consistent with the real situation. At this time, the expression of the time–frequency power spectral density can be written as Formula (5):

$$G_q(f) = G_q(n_0){n_0}^2\frac{v}{f^2 + {f_{min}}^2v^2} \tag{5}$$

In Formula (5), $f_{min}$—time lower limit cut-off frequency.

The relationship between the power spectral density of the excitation quantity and the power spectral density of the response quantity of the first-order filtered white noise system is expressed as follows:

$$G_q(f) = |H(w)|^2 S(f) \tag{6}$$

In Formula (6), *S(f)*—power spectral density of the excitation quantity and *H(w)*—power spectral density of response quantity [29].

$$q(f) = H(w)S(f) \tag{7}$$

In Formula (7), *q(f)*—frequency domain representation of the response quantity.
The time domain expression of the first-order filtered white noise system is:

$$A\dot{q}(t) + Bq(t) = CS(t) \tag{8}$$

In the Formula, *S(t)* is the white noise signal, A, B and C are system parameters, $\dot{q}(t)$ is the time domain representation of the response quantity. The standard white noise *S(t)* is used as the excitation input to the first-order filtered white noise system represented by Equation (8), and the response quantity *q(t)* can be used as pavement roughness [32]. After Fourier transform and derivation, the random pavement model can be expressed by the following Formula [33]:

$$\dot{q}(t) + 2\pi f_{min}q(t) = 2\pi n_0 \sqrt{G_q(n_0)v}S(t) \tag{9}$$

3.1.2. Construction of Random Signal Pavement Model

Based on Equation (9), the model can be established in MATLAB Simulink. Figure 5 shows the random pavement model based on filtered white noise.

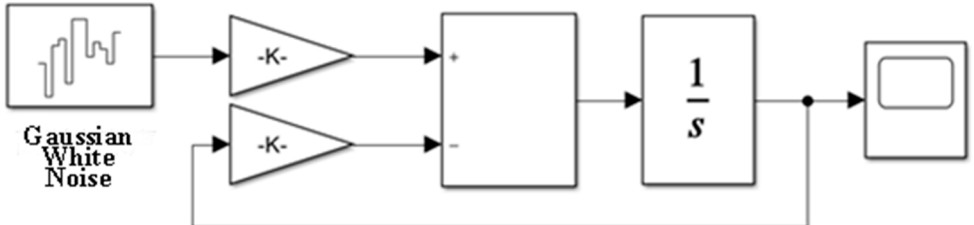

**Figure 5.** Random pavement model based on filtered white noise.

Referring to Table 3, the relatively flat class A road surface and the slightly fluctuating class C road surface were selected for simulation at speeds of 30 km/h and 60 km/h. The different working conditions are shown in Table 4.

**Table 4.** Simulated working conditions.

| Working Condition | Description |
| --- | --- |
| Condition 1 | Class A pavement, 30 km/h |
| Condition 2 | Class A pavement, 60 km/h |
| Condition 3 | Class C pavement, 30 km/h |
| Condition 4 | Class C pavement, 60 km/h |

Bringing the required parameters into Equation (9) and running the simulation in Simulink, we obtained the road roughness diagram under different speeds.

From Figure 6, it can be seen that when the pavement grade is the same, and the speed is positively correlated with the pavement roughness, that is, the higher the speed, the greater the road roughness. At the same speed, the pavement grade was positively

correlated with pavement roughness. The higher the pavement grade, the greater the roughness of the pavement.

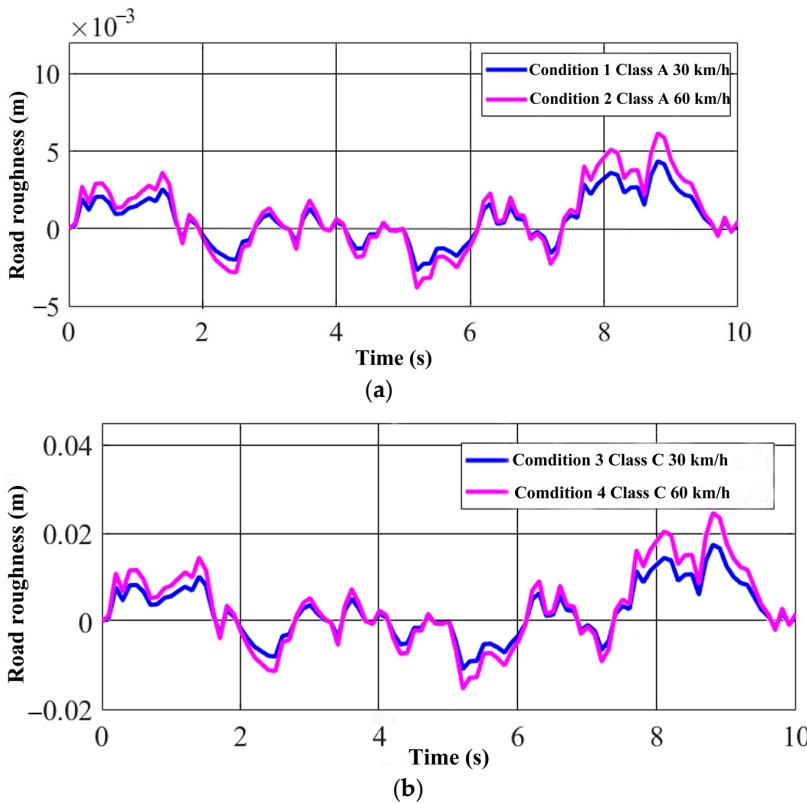

**Figure 6.** (**a**) Road roughness curve under condition 1 (Class A pavement, 30 km/h) and condition 2 (Class A pavement, 60 km/h); (**b**) Road roughness curve under condition 3 (Class C pavement, 30 km/h) and condition 4 (Class C pavement, 60 km/h).

### 3.2. Establishment of Air Suspension Model

3.2.1. Modeling of Single-Axle Four-Airbag Passive Air Suspension System

The single-axle four-airbag passive air suspension system model is shown in the following figure:

The single-axle four-airbag active air suspension system, based on the passive suspension shown in Figure 7, incorporates an additional force that can be adjusted in size and direction at any time to counteract the vibrations induced by the road surface on the vehicle body.

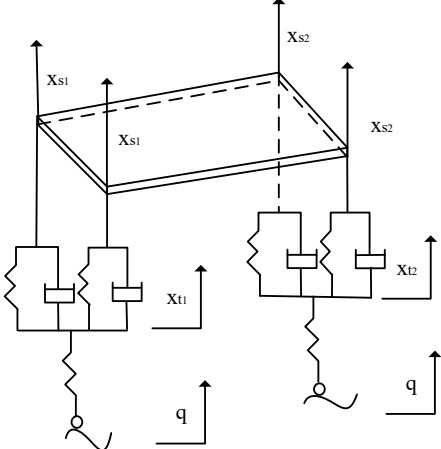

**Figure 7.** Single-axle four-airbag passive air suspension system model.

### 3.2.2. Modeling of 1/4 Active Air Suspension System

For ease of study, the following assumptions have been made: (1) The active air spring model ignores the roll effect of the single-axle four-airbag suspension and the torsional force generated by the single-side air spring connected by the bracket. (2) The two air springs connected by the bracket are symmetric, so they can be simplified into one air spring. (3) The left and right air suspension systems of heavy vehicles are entirely symmetrical, and the coupling effect of the front and rear air suspension systems is ignored. (4) The tire and the ground are assumed to have no sliding and maintain good contact. (5) The vehicle's tires are treated as springs without considering damping in the model. The two-degree-of-freedom 1/4 vehicle active air suspension system model was established, as follows.

In Figure 8, $m_s$ is the sprung mass, $m_t$ is the unsprung mass, $F_a$ is the active control force of the controller, $k_t$ is the tire stiffness, $k_s$ is the stiffness of the suspension system, $c_s$ is the damping coefficient of the suspension system, $q$ is the road displacement function, $x_t$ is the vertical displacement of the equivalent unsprung mass of the suspension, $x_s$ is the vertical displacement of the unsprung mass of the suspension system, and $x_r$ is the pavement displacement.

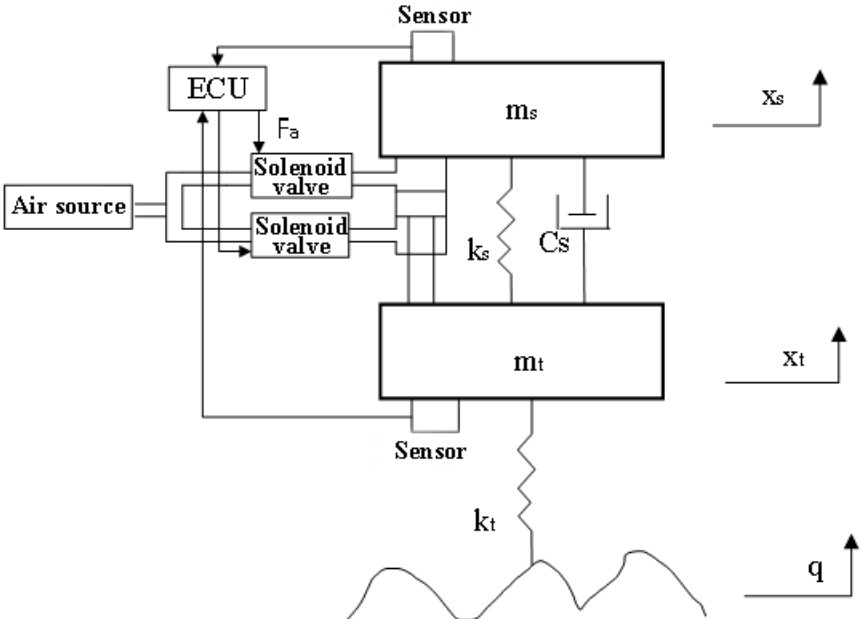

**Figure 8.** Two-degree-of-freedom 1/4 vehicle active air suspension system model.

Taking $m_s$ and $m_t$ as the research objects, respectively, the vibration differential equation of the two degree of freedom active air suspension system model can be established using Newton's second law, as shown in the Equations (10) and (11):

$$m_s\ddot{x}_s + c_s(\dot{x}_s - \dot{x}_t) + k_s(x_s - x_t) + F_a = 0 \tag{10}$$

$$m_t\ddot{x}_t - c_s(\dot{x}_s - \dot{x}_t) + k_s(x_s - x_t) + k_t(x_t - x_r) - F_a = 0 \tag{11}$$

The vibration differential equation of the passive suspension system has no active control force, and the differential equation is as follows:

$$m_s\ddot{x}_s + c_s(\dot{x}_s - \dot{x}_t) + k_s(x_s - x_t) = 0 \tag{12}$$

$$m_t\ddot{x}_t - c_s(\dot{x}_s - \dot{x}_t) + k_s(x_s - x_t) + k_t(x_t - x_r) = 0 \tag{13}$$

According to the vibration differential equation of the air suspension system model, the sprung mass was built in MATLAB Simulink. The block diagram is shown in Figure 9.

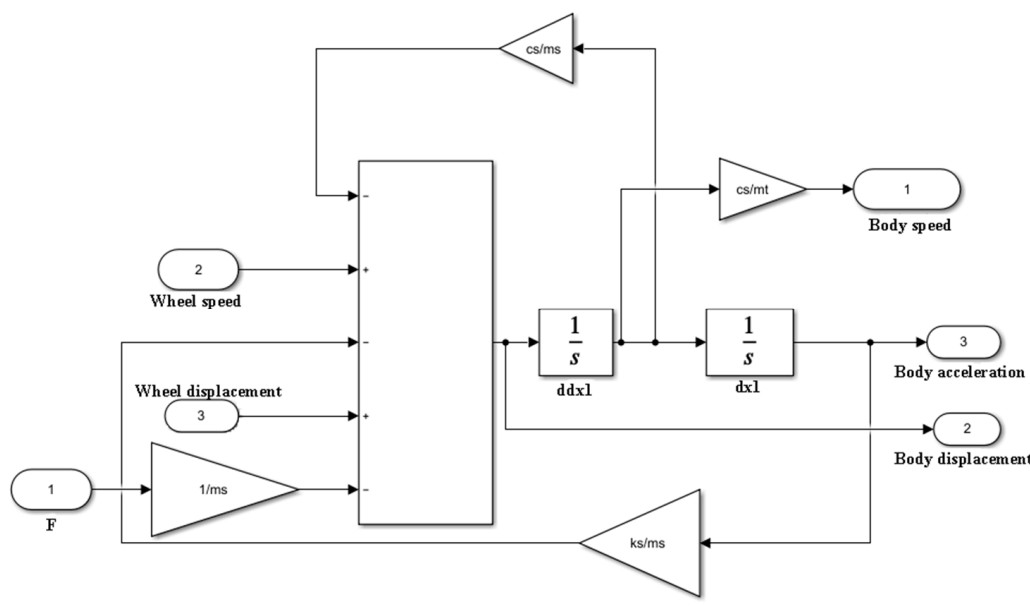

**Figure 9.** Sprung mass simulation block diagram.

According to the vibration differential equation of the air suspension system model, the unsprung mass was built in MATLAB Simulink. The block diagram is shown in Figure 10.

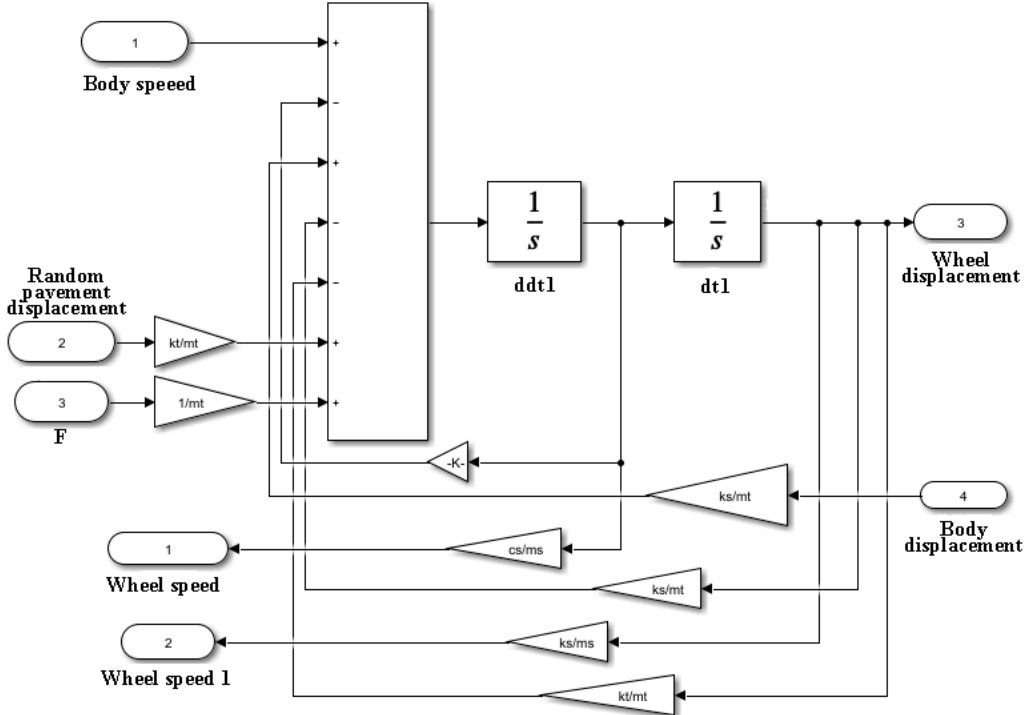

**Figure 10.** Unsprung mass simulation block diagram.

The block diagram of the passive suspension system built in MATLAB according to the differential equation is shown in Figure 11.

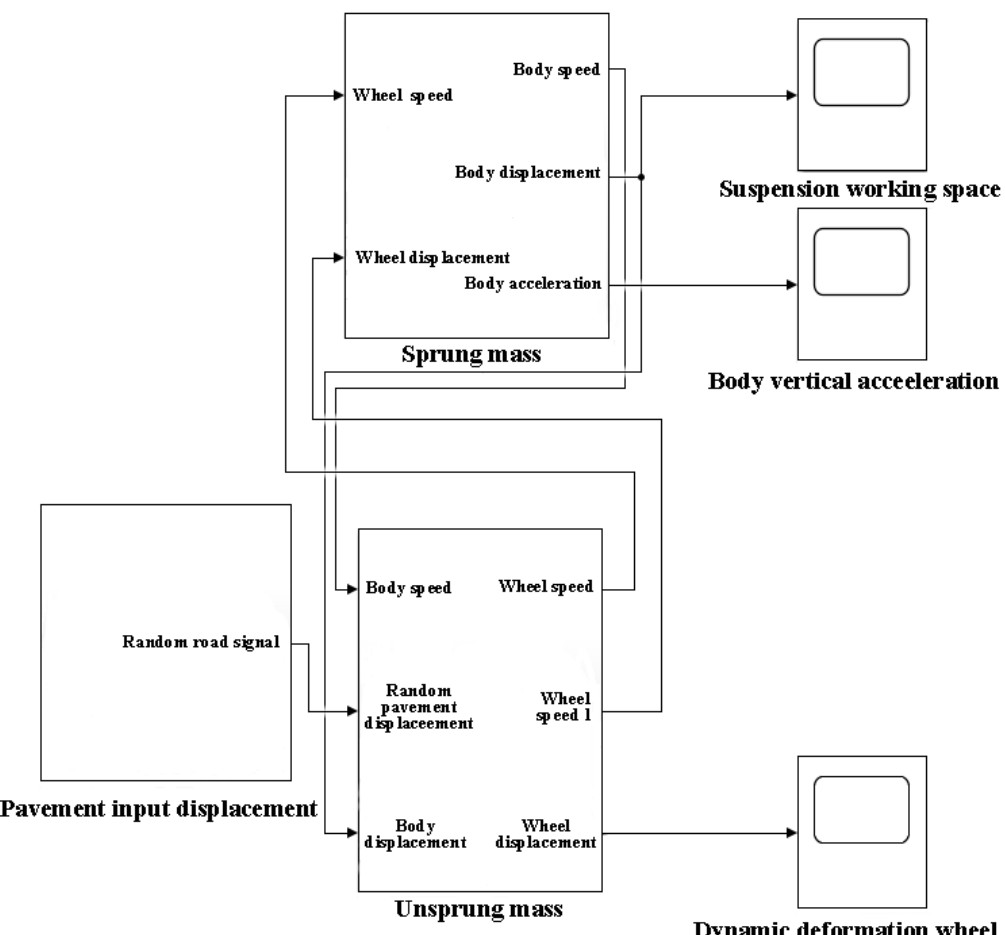

**Figure 11.** Simulation block diagram of passive suspension system.

### 3.3. Design of Fuzzy PID Controller

In this study, a fuzzy PID controller was selected as the control strategy for the air suspension system. The fuzzy PID controller is an extension of the traditional PID control algorithm with the addition of a fuzzy controller. Compared to the conventional PID controller, which requires manual adjustment of individual parameters, the fuzzy PID controller can automatically adjust parameters based on the input error (E) and error change rate (EC) from the fuzzy controller. This results in time savings and efficient parameter tuning, leading to improved control precision.

In the fuzzy PID controller, the error (E) represents the difference between the desired value (output) and the actual value (input), while the error change rate (EC) signifies the rate of change of the error. Depending on the values of the error and error change rate, the three parameters of the fuzzy PID controller ($K_p$, $K_i$, and $K_d$) are dynamically adjusted using specific fuzzy control rules.

#### 3.3.1. Fuzzy Quantization of Controller Parameters

There are two inputs in the fuzzy controller, error E and error change rate EC, which are described by seven fuzzy language subsets, namely {negative large, negative medium, negative small, zero, positive small, positive medium and positive large}, and the definitions are given:

$$E = \{NB, NM, NS, Z, PS, PM, PB\}$$

$$EC = \{NB, NM, NS, Z, PS, PM, PB\}$$

There are three outputs in the fuzzy controller, which are also described by seven fuzzy language subsets, and the definitions are:

$$\triangle K_p = \{NB,NM,NS,Z,PS,PM,PB\}$$

$$\triangle K_i = \{NB,NM,NS,Z,PS,PM,PB\}$$

$$\triangle K_d = \{NB,NM,NS,Z,PS,PM,PB\}$$

The variable universe is [−1, 1], and the universe is divided according to the rule: half interval width = (total length of universe/12) × 2. That is, the half interval length is 0.333.

### 3.3.2. Establishment of Fuzzy PID Membership Function

Through the fuzzy logic controller in the library browser option in Simulink, we built a fuzzy controller module in Simulink, as shown in Figure 12. We entered "fuzzy" in the MATLAB 2020a command line window and executed it to obtain a "fuzzy logic designer" and design of a two-input three-output system.

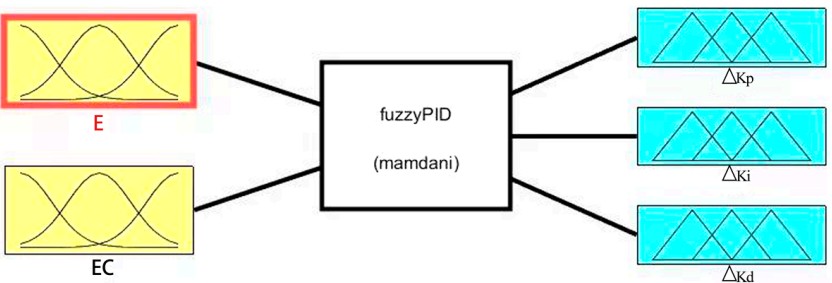

**Figure 12.** Fuzzy controller module.

We set $\Delta k_p$, $\Delta k_i$, and $\Delta k_d$, and defined the membership functions for two input variables and three output variables to obtain the following figures: Figure 13 shows the membership function of input variables, and Figure 14 shows the membership function of output variables.

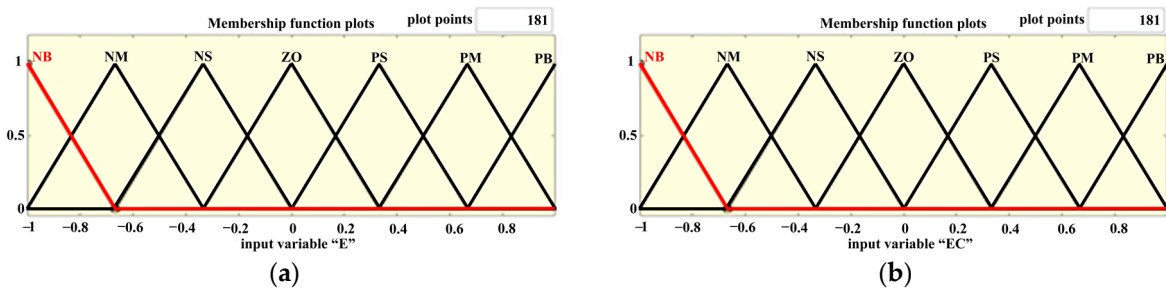

**Figure 13.** (**a**) Membership function of input variable E; (**b**) Membership function of input variable EC.

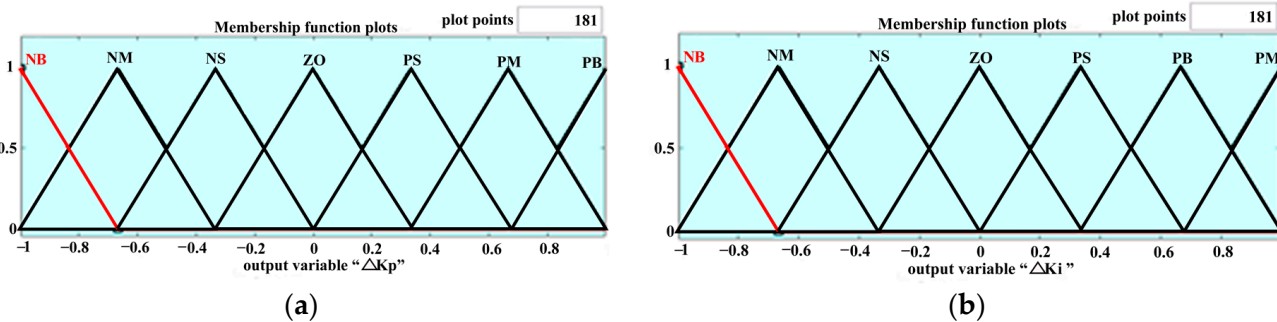

**Figure 14.** *Cont.*

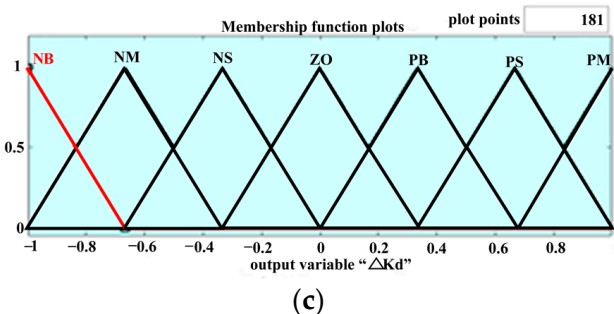

(**c**)

**Figure 14.** (**a**) Membership function of output variable $\Delta k_p$; (**b**) Membership function of output variable $\Delta k_i$; (**c**) Membership function of output variable $\Delta k_d$.

### 3.3.3. Design of Fuzzy PID Controller

The principles for tuning the fuzzy PID parameters are as follows: (1) When the absolute value of E is relatively large, indicating that the system is in the response stage, the response speed should be accelerated. To achieve this, the $K_p$ value is increased, the $K_i$ value is decreased, and to avoid differential saturation and excessive overshoot in the system response, the $K_d$ value is set to 0. (2) When the product of E and EC is greater than 0, it indicates that the absolute value of the system state error is continuously increasing. When the absolute value of E is not significantly different from the absolute value of EC, the system is in the tracking stage. To reduce system overshoot, the values of $K_p$, $K_i$, and $K_d$ should not be too large. When the absolute value of E is relatively large, the $K_p$ value can be appropriately increased, the $K_i$ value decreased, and the $K_d$ value set to a medium level to ensure the system's dynamic and steady-state performance. When the absolute value of E is small, a large $K_d$ value, a large $K_i$ value, and a medium $K_p$ value should be used to prevent system oscillations. (3) When the product of E and EC is less than 0, it indicates that the error is decreasing. When the absolute value of E is relatively large, to ensure the system's dynamic and steady-state performance, a medium $K_p$ value, a smaller $K_i$ value, and a medium $K_d$ value should be selected. When the absolute value of E is small, to maintain good steady-state performance, both $K_p$ and $K_i$ values should be increased, and a medium $K_d$ value is appropriate.

Based on the above principles and combined with the control experience of the active air suspension system, the fuzzy relationship between the input and output variables of the fuzzy PID controller was determined. The control rules shown in Tables 5–7 such as $\Delta K_p$, $\Delta K_i$, and $\Delta K_d$, are mainly derived from fuzzy control theory combined with expert experience.

**Table 5.** $\Delta k_p$ fuzzy rule table.

| c \ ec | NB | NM | NS | Z | PS | PM | PB |
|---|---|---|---|---|---|---|---|
| NB | NB | NB | NM | NM | NS | Z | Z |
| NM | NB | NB | NM | NS | NS | Z | PS |
| NS | NM | NM | NS | NS | Z | Z | PS |
| Z | NM | NM | NS | Z | PS | PS | PM |
| PS | NS | NS | Z | PS | PS | PS | PM |
| PM | NS | Z | PS | PS | PM | PM | PB |
| PB | Z | Z | PM | PM | PM | PB | PB |

MATLAB will automatically perform the defuzzification process using the Center of Gravity (COG) method through its internal program. By opening the Surface Viewer, the

membership functions determined by the control rules can be observed, as shown in Figure 15.

**Table 6.** $\Delta k_i$ fuzzy rule table.

| ec \ c | NB | NM | NS | Z | PS | PM | PB |
|---|---|---|---|---|---|---|---|
| NB | NB | NB | NM | NS | NS | Z | NS |
| NM | NB | NB | NM | NS | NS | Z | NS |
| NS | NM | NM | NS | Z | PS | PS | PS |
| Z | NM | NM | NS | Z | PS | PS | PM |
| PS | NM | NS | Z | PS | PS | PM | PB |
| PM | Z | Z | PS | PS | PM | PM | PB |
| PB | Z | Z | PS | PS | PM | PM | PB |

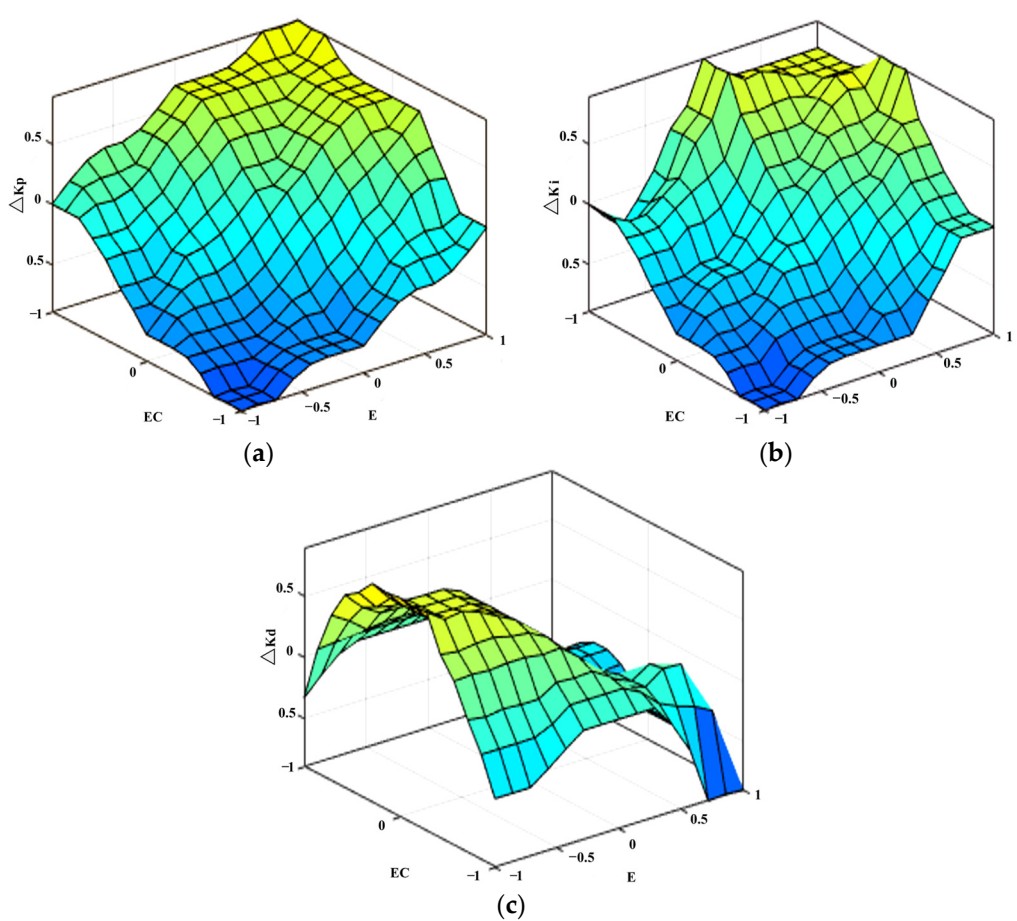

**Figure 15.** (**a**) Observation diagram of the membership function of $\Delta K_p$; (**b**) Observation diagram of the membership function of $\Delta K_i$; (**c**) Observation diagram of the membership function of $\Delta K_d$.

**Table 7.** $\Delta k_d$ fuzzy rule table.

| ec \ c | NB | NM | NS | Z | PS | PM | PB |
|---|---|---|---|---|---|---|---|
| NB | NS | NS | PB | PB | PM | PM | Z |
| NM | NS | NS | PS | PS | PM | Z | Z |
| NS | Z | Z | PS | PM | PM | PS | Z |
| Z | Z | Z |  |  |  |  |  |

**Table 7.** *Cont.*

| ec \ c | NB | NM | NS | Z | PS | PM | PB |
|---|---|---|---|---|---|---|---|
| PS | | | | | | | |
| PM | NB | | PS | | | NS | NB |
| PB | NB | | NM | | | NS | NB |

After setting the fuzzy control rules, the scaling factors for two input variables and the quantization factors for three output variables are calculated. The scaling factor for "e" is 0.25 and 0.005 for "ec". The quantization factors for the three outputs are 20, 10, and 0.6, respectively. The proportional control parameter for the control system is set to 1000, the integral control parameter to 8000, and the differential control parameter to 0.2. By adjusting the control parameters of each control link, the controller's control effect can be changed, and the control quality can be optimized [34]. Based on the fuzzy PID control principle, the fuzzy PID controller can be obtained as shown in Figure 16:

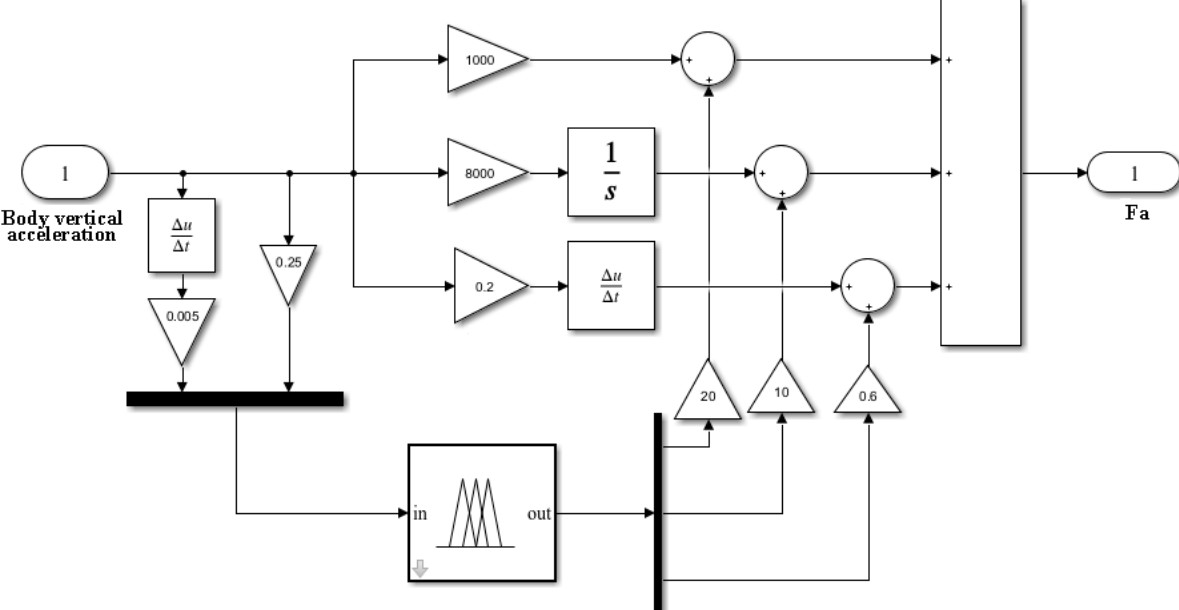

**Figure 16.** Fuzzy PID controller.

Figure 16 depicts that the input of the fuzzy PID controller is the error e, which represents the difference between the system output and input values. In this case, the system's output value is the vertical acceleration of the vehicle body, while the expected value serves as the system input value. As the desired value, the ideal vertical acceleration of the vehicle body is 0. Therefore, the error e is also equivalent to the vertical acceleration of the vehicle body. The output of the fuzzy PID controller system is the active control force Fa.

Based on the differential equations, a block diagram of the active suspension system was constructed in Matlab. The main components of the active air suspension system include the vehicle body, tire part, road input part, and the fuzzy PID controller. The simulation diagram of the 1/4 active air suspension system is shown in Figure 17.

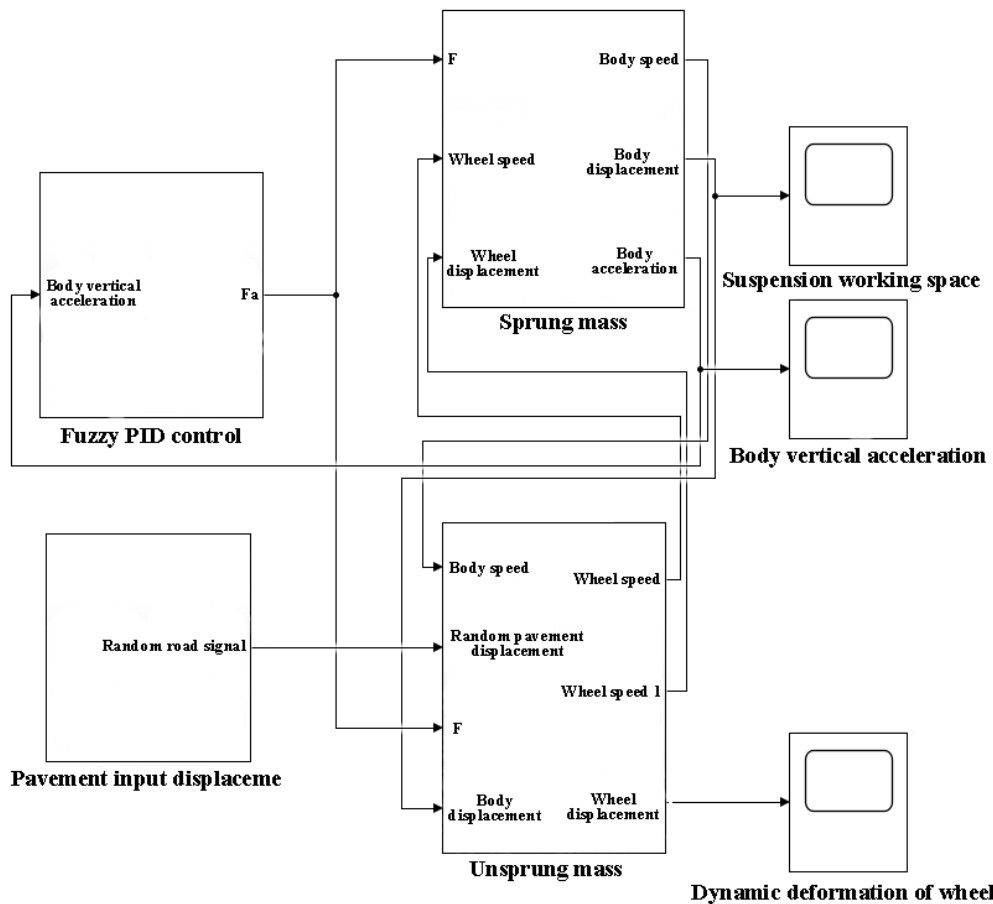

**Figure 17.** Simulation block diagram of active suspension system.

## 4. Simulation Results and Analysis

The simulation parameters of the 1/4 active air suspension system for a heavy vehicle can be obtained by derivation, as shown in Table 8.

**Table 8.** 1/4 active air suspension system simulation parameters.

| Name | Numerical Value | Unit |
|---|---|---|
| Sprung mass | 5000 | Kg |
| Unsprung mass | 602 | Kg |
| Suspension vertical stiffness | 241,200 | N/m |
| Wheel stiffness | 192,000 | N/m |
| Suspension damping | 6875.505 | N $\times$ s/m |

There are three primary indicators used to assess the vibration characteristics of two-degree-of-freedom active air suspension systems: Sprung Mass Acceleration (SMA), Dynamic Tire Load (DTL), and Suspension Working Space (SWS).

(1) SMA, which is the difference between the vertical body acceleration and the wheel acceleration, is a crucial parameter that measures the smoothness of heavy vehicle driving and can be utilized to evaluate the driving performance and comfort.

(2) DTL, which is the difference between body displacement and wheel displacement, is a significant indicator of the handling stability of heavy vehicles and can reflect the body attitude. In simulation analyses, it is usually represented by Dynamic Tire Deformation (DTD).

(3)    SWS, which is the difference between the wheel displacement and the road input displacement, is a crucial index used to evaluate the grounding stability of heavy vehicles [35].

After building the model using the MATLAB Simulink integration module, the curves for the body acceleration SMA, suspension dynamic travel DTD, and wheel dynamic deformation SWS of the heavy-duty vehicle passive suspension and the heavy-duty vehicle active air suspension system were obtained using an oscilloscope. Figure 18 shows the time-domain comparison of vibration curves for the air suspension system of heavy-duty vehicles on Grade A pavement at 30 km/h. Figure 19 shows the time-domain comparison of vibration curves for the air suspension system of heavy-duty vehicles on Grade A pavement at 60 km/h. Figure 20 shows the time-domain comparison of vibration curves for the air suspension system of heavy-duty vehicles on Grade C pavement at 30 km/h. Figure 21 shows the time-domain comparison of vibration curves for the air suspension system of heavy-duty vehicles on Grade C pavement at 60 km/h.

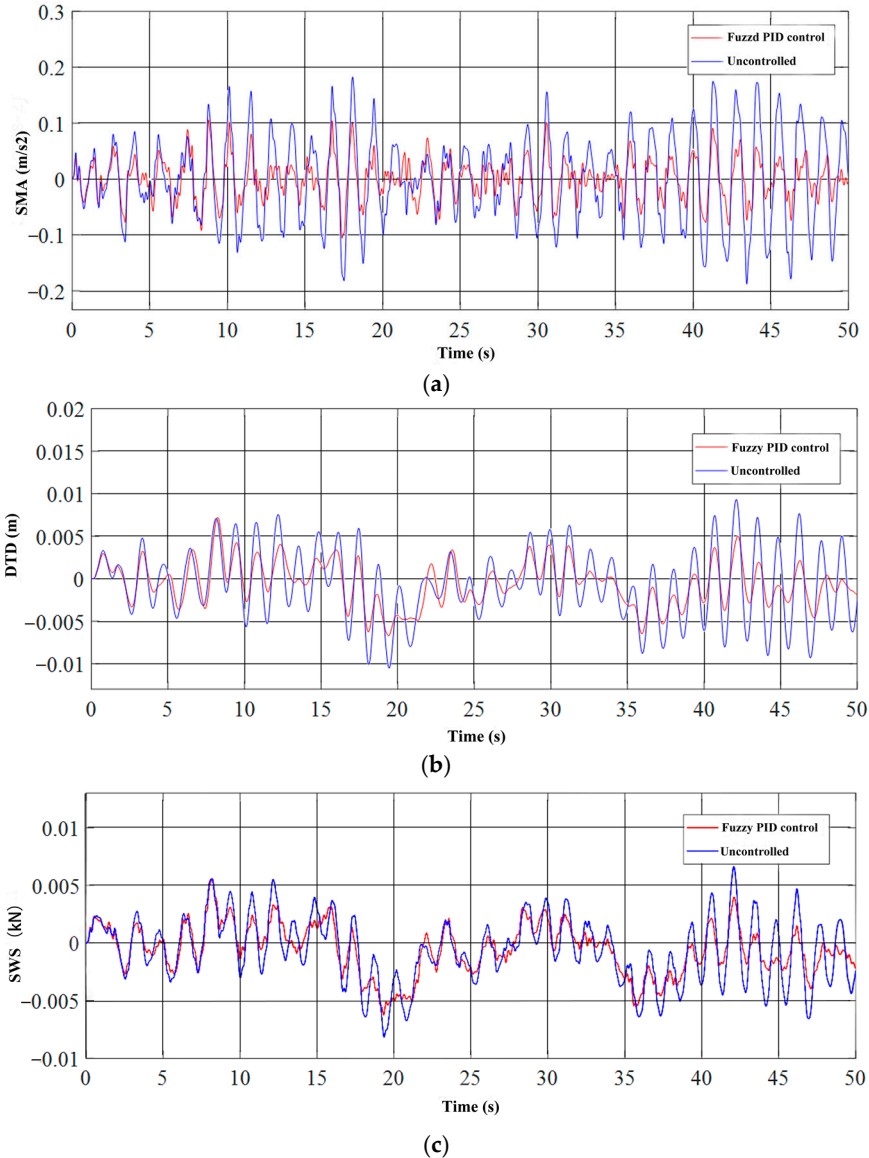

**Figure 18.** Time-domain comparison of vibration curves for the air suspension system of heavy-duty vehicles on Grade A pavement at 30 km/h. (**a**) Body acceleration SMA. (**b**) Suspension dynamic travel DTD. (**c**) Tire dynamic load SWS.

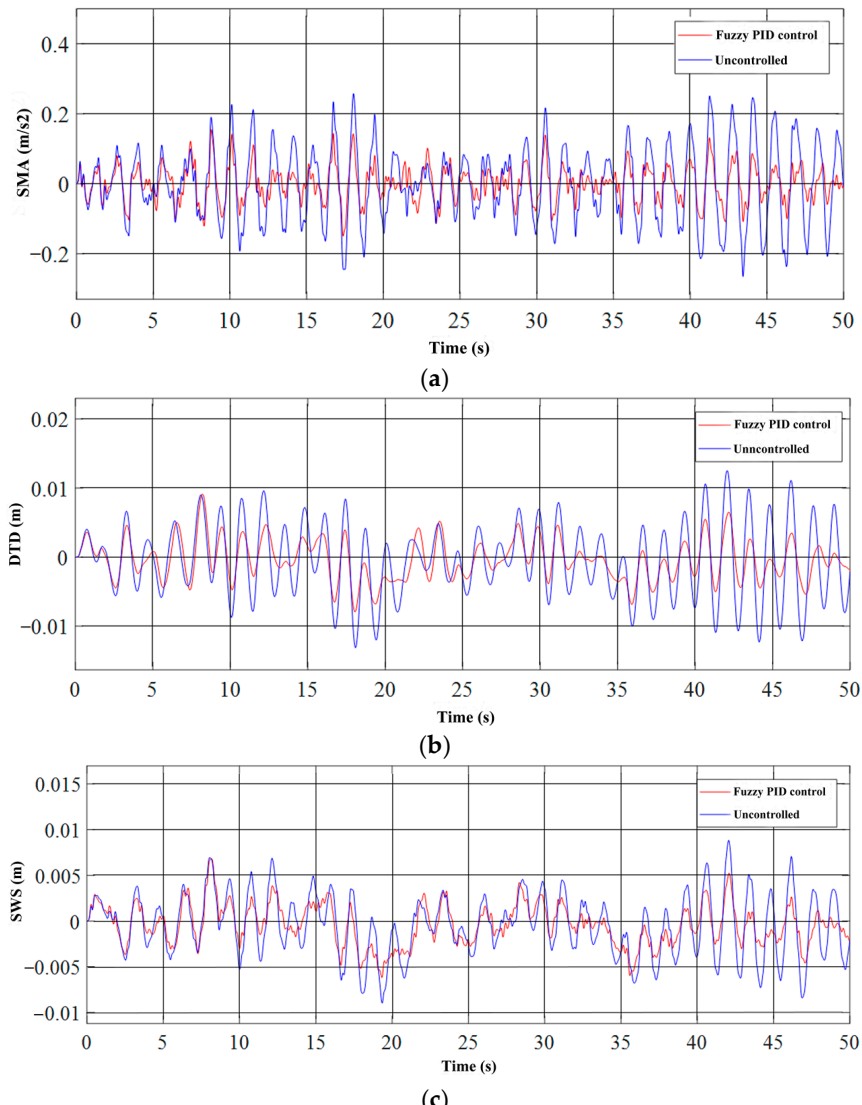

**Figure 19.** Time-domain comparison of vibration curves for the air suspension system of heavy-duty vehicles on Grade A pavement at 60 km/h. (**a**) Body acceleration SMA. (**b**) Suspension dynamic travel DTD. (**c**) Tire dynamic load SWS.

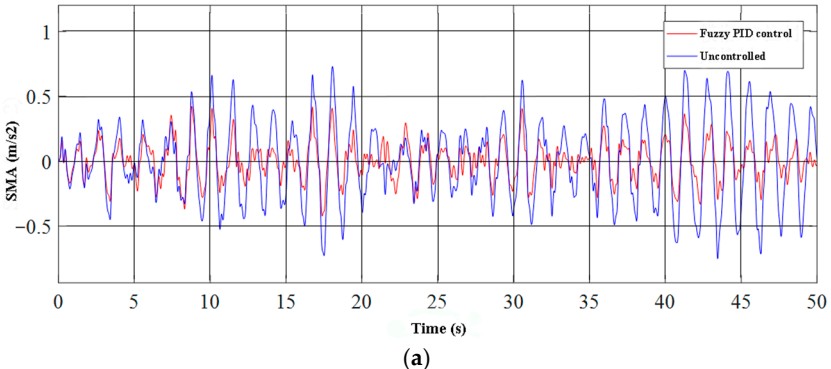

**Figure 20.** *Cont.*

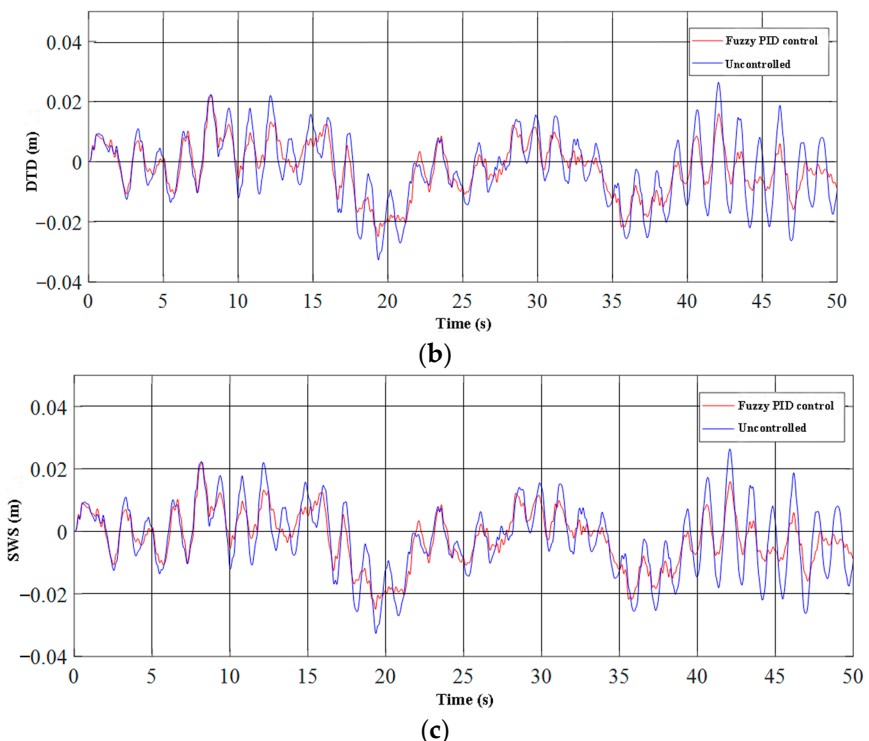

**Figure 20.** Time-domain comparison of vibration curves for the air suspension system of heavy-duty vehicles on Grade C pavement at 30 km/h. (**a**) Body acceleration SMA. (**b**) Suspension dynamic travel DTD. (**c**) Tire dynamic load SWS.

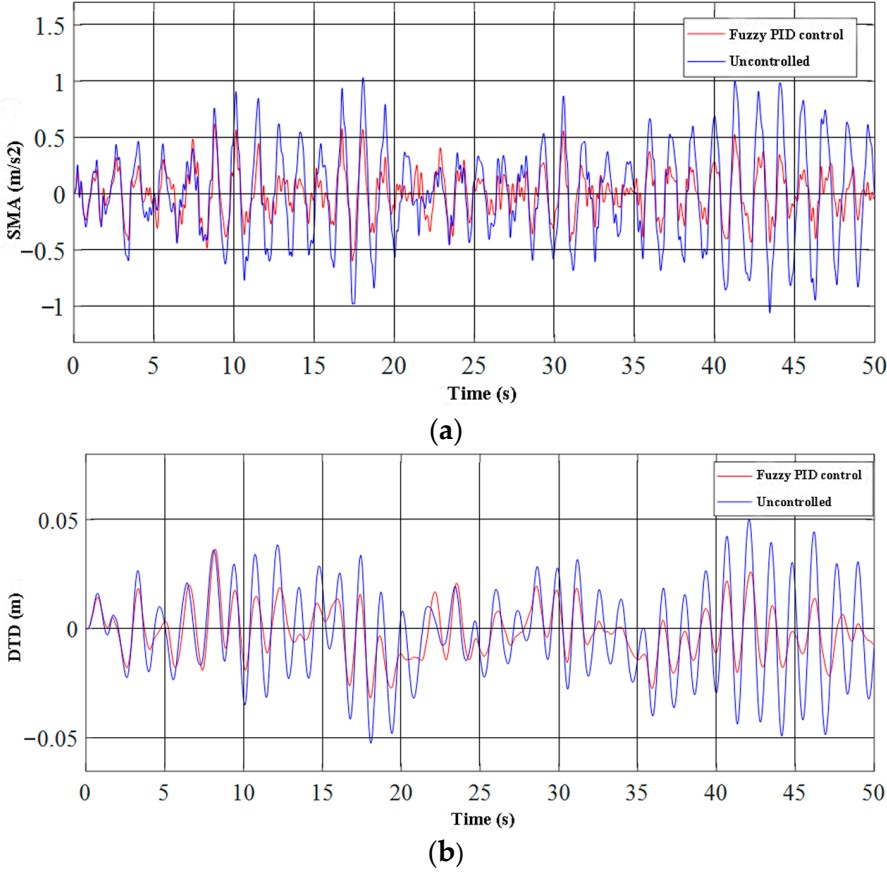

**Figure 21.** *Cont.*

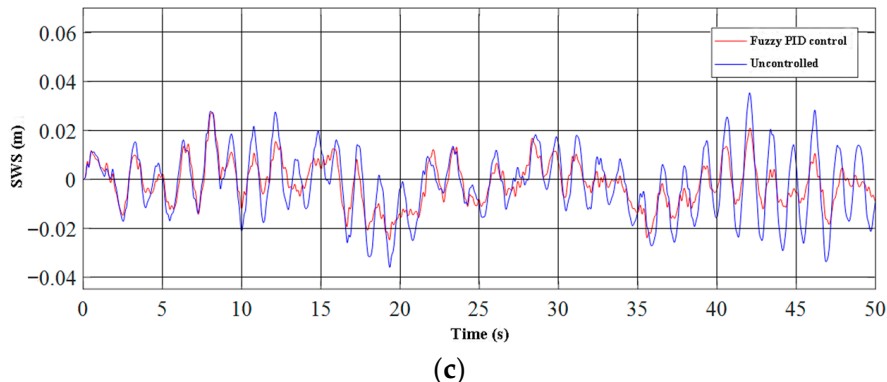

**Figure 21.** Time-domain comparison of vibration curves for the air suspension system of heavy-duty vehicles on Grade C pavement at 60 km/h. (**a**) Body acceleration SMA. (**b**) Suspension dynamic travel DTD. (**c**) Tire dynamic load SWS.

After comparing Figure 18a with Figure 19a, and Figure 20a with Figure 21a, it can be concluded that the higher the vehicle speed, the higher the body vertical acceleration SMA. By comparing Figure 18b with Figure 19b, and Figure 20b with Figure 21b, it can be concluded that the higher the vehicle speed, the higher the suspension dynamic travel DTD. By comparing Figure 20c with Figure 21c, it can be concluded that the higher the vehicle speed, the higher the dynamic tire load SWS. It can be seen that the dynamic tire load SWS also increases with speed by comparing Figure 18c with Figures 19c and 20c with Figure 21c.

Moreover, it can be found that the fuzzy PID control has obvious superiority over the uncontrolled air suspension system at different speeds on the same level of road surface. The body vertical acceleration SMA, suspension dynamic travel DTD, and tire dynamic load SWS indexes were all improved with the fuzzy PID control. The improvement of the body acceleration SMA was the most significant, followed by suspension dynamic travel DTD, while the improvement of the dynamic tire load SWS was relatively smaller. In summary, by comparing the time domain curves at different speeds on a C-grade road surface, it can be concluded that the fuzzy PID control can effectively mitigate the vibration situation of heavy vehicles during driving.

The root mean square values of the characteristics obtained from the simulation results of the heavy vehicle air suspension system are presented in Table 9.

**Table 9.** Statistical characteristics of passive and active air suspension in the simulation.

| RMS | Pavement Grade | Vehicle Speed | Fuzzy PID Control | Uncontrolled | Improvement Rate |
|---|---|---|---|---|---|
| | A | 30 km/h | $0.03950 \text{ m/s}^{-2}$ | $0.05082 \text{ m/s}^{-2}$ | 22.3% |
| Body vertical acceleration (SMA) | A | 60 km/h | $0.05466 \text{ m/s}^{-2}$ | $0.0702 \text{ m/s}^{-2}$ | 22.1% |
| | C | 30 km/h | $0.1580 \text{ m/s}^{-2}$ | $0.2033 \text{ m/s}^{-2}$ | 22.3% |
| | C | 60 km/h | $0.2186 \text{ m/s}^{-2}$ | $0.2808 \text{ m/s}^{-2}$ | 22.2% |
| | A | 30 km/h | 0.002424 m | 0.002951 m | 17.9% |
| Suspension dynamic travel (DTD) | A | 60 km/h | 0.003050 m | 0.003821 m | 20.1% |
| | C | 30 km/h | 0.009695 m | 0.01180 m | 17.8% |
| | C | 60 km/h | 0.01220 m | 0.01528 m | 20.1% |
| | A | 30 km/h | 0.001957 m | 0.002186 m | 10.5% |
| Tire dynamic load (SWS) | A | 60 km/h | 0.002363 m | 0.002707 m | 12.7% |
| | C | 30 km/h | 0.007828 m | 0.008745 m | 10.5% |
| | C | 60 km/h | 0.009453 m | 0.010833 m | 11.9% |

By analyzing the vibration time-domain curves and data of the heavy vehicle air suspension system, it is evident that the fuzzy PID control provides clear advantages under different road conditions and speeds. When compared to the air suspension system without control, the fuzzy PID control active air suspension system showed improvements in the body vertical acceleration SMA, suspension dynamic stroke DTD, and tire dynamic load SWS indexes. The most significant improvement was observed when the vehicle speed on a grade A road was 60 km/h, with a reduction of 22.1% in the vertical acceleration of the vehicle body, 20.1% in the dynamic travel of the suspension, and 12.7% in the dynamic load of the tire. Therefore, it can be concluded that the fuzzy PID active air suspension system can effectively reduce vehicle vibration and maintain a certain degree of comfort.

## 5. Conclusions

Based on the structural specifications of a 6 × 4 heavy-duty vehicle and the desired performance of an active air suspension system, a double-axle eight-airbag active air suspension system was designed. The system utilizes the fundamental properties of air springs to achieve intelligent height adjustment that can adapt to various road conditions.

A random road excitation model was established to simulate different road conditions, and a 1/4 active air suspension system model was designed for a double-axle eight-airbag active air suspension system. Two types of road conditions were selected, a relatively flat class A road and a slightly fluctuating class C road, and two different vehicle speeds, 30 km/h and 60 km/h, were tested. By using the MATLAB Simulink simulation platform, a fuzzy PID controller was designed and the block diagrams of 1/4 active air suspension system and 1/4 passive air suspension system were built. The evaluation indexes used were the vertical acceleration of the body, the dynamic travel of the suspension, and the dynamic deformation of the tire. Comparisons of the results showed that the fuzzy PID control active air suspension system outperformed the passive air suspension system in all three indexes. In particular, when the vehicle speed on a class A road was 60 km/h, the vertical acceleration of the vehicle body was reduced by 22.1%, the dynamic travel of the suspension was reduced by 20.1%, and the dynamic load of the tire was reduced by 12.7%.

**Author Contributions:** Conceptualization, X.B. and L.L.; methodology, L.L.; software, X.B.; validation, C.Z. and W.G.; formal analysis, X.B.; investigation, C.Z.; resources, L.L.; data curation, W.G.; writing—original draft preparation, X.B.; writing—review and editing, X.B.; visualization, L.L.; supervision, X.B.; project administration, L.L. All authors have read and agreed to the published version of the manuscript.

**Funding:** This research received no external funding.

**Institutional Review Board Statement:** Not applicable.

**Informed Consent Statement:** Not applicable.

**Data Availability Statement:** Not applicable.

**Conflicts of Interest:** The authors declare no conflict of interest.

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
