# Peer review of "Research on Height Adjustment Characteristics of Heavy Vehicle Active Air Suspension Based on Fuzzy Control"

_wevj, doi:10.3390/wevj14080210_

Round 1

Reviewer 1 Report

This paper focuses on simulating and evaluating the performance of the air suspension system. This is an exciting topic; however, this manuscript is not well written. The authors should try to improve more.

1) The description "Additionally, the active air suspension system enables the vehicle to achieve lightweight, automation and intelligence" is inaccurate. Vehicles using air suspension systems often have a huge mass. The authors should elaborate on this statement in more detail.

2) Authors should detail analyze and evaluate references rather than just listing them, such as [14-17] and [18-23].

3) Authors should analyze and show the difference between active and air suspensions. In addition, the authors should add some references to improve the quality of the literature review, including:

10.1016/j.heliyon.2023.e14210

10.1038/s41598-022-24069-w

10.1590/1679-78256621

4) The paper's layout should be indicated at the end of Section 1.

5) The values in equation (9) should be explained clearly.

6) The unit of time in Figure 6b is incorrect.

7) In Figure 12, what are the symbols E and EC?

8) Why do the authors use Fuzzy PID controllers?

9) What is the basis for choosing fuzzy rules (Table 5)?

10) Simulation results should show the change in sprung mass displacement and acceleration.

11) Reference [26] has not been cited in the manuscript. Additionally, Figure 2, Figure 5, Figure 9, Figure 10, Figure 11, Figure 12, Figure 13, Figure 14, Figure 15, Figure 17, Figure 18, Figure 19, Figure 20, and Figure 21 are not fully cited.

This should be improved.

Reviewer 2 Report

This study focuses on improving heavy vehicle suspension systems by designing an active air suspension system for a specific 6x4 heavy vehicle. The active air suspension adjusts stiffness, damping, and body height in real time based on road conditions, resulting in a smoother ride and more efficient vehicle loading/unloading. The following comments may be of help to improve its completeness and eligibility for publication.    

The following comments may be of help to improve its completeness and eligibility for publication.    

1 - Explain more about the novelty of the article at the end of the introduction.

2 - The graphic quality of the figures used in the manuscript could be improved and uniformed since some of them are difficult to read (text size, subtitles, legends). Remove the gray background from the figures.

3 - Improve the resolution of Figure 4. Place its legend on the same page as the Figure. Check the text indicating the components in the schematic diagram. The text Altitude sensor 1, Solenoid valve, and others overlap the lines.

4 – “Where, v——vehicle speed, unit: m*s-1.” Remove the asterisk indicating multiplication.

5 – Improve the quality of Figure 6. Add a grid to the graph. Put the unit of measurement in brackets, do not use /m. Correct the caption of Figure 6b "time/m".  Align Figures 6a and 6b.

6 – Air springs have non-linear characteristics, however, the authors use a linear model. Explain the reason for the simplification in the model assumptions.

7 – “There are two inputs in the fuzzy controller, error e and error change rate ec…” . Explain how this error is determined or what it means.

8 – Put all of Figure 14 on a single page. Same for Table 7.

9 – Increase the text in Figure 17.

10 – Indicate which defuzzyfication method is used in the MatLab toolbox.

11 – Determine parameters of interest for the suspension, such as natural frequency, damped natural frequency, and damping factor. Add analysis of the air suspension in the frequency domain, fuzzy PID control, and uncontrolled, indicating its natural frequencies.  

12 – Standardize the graphs in Figures 18, 19, 20, and 21.  Figure 18 c and Figure 21 c are different from the others. For all figures, improve resolution, and add a grid to the graph.

A careful revision of the text would be interesting, some passages are difficult to understand.

Round 2

Reviewer 1 Report

Regarding concern #11, the authors should mention the location where these Figures appear in this article. For example, Figure 2 shows vehicle parameters of heavy vehicles; Figure 5 gives information about random pavement, etc.

No

Author Response

Dear Reviewer,

Thank you very much for your valuable feedback and guidance. Based on your suggestions, we have annotated the figures in the paper to make it clearer and more understandable. We believe that these modifications will enhance the quality and readability of the paper. Once again, we appreciate your valuable input, and your professional guidance has been instrumental in our research work.

Thank you!

Reviewer 2 Report

This study focuses on improving heavy vehicle suspension systems by designing an active air suspension system for a specific 6x4 heavy vehicle. The active air suspension adjusts stiffness, damping, and body height in real time based on road conditions, resulting in a smoother ride and more efficient vehicle loading/unloading. The following comments may be of help to improve its completeness and eligibility for publication. 

I am satisfied with the corrections made by the authors

Author Response

Dear Reviewer,

Thank you very much for expressing your satisfaction with the corrections we have made. We sincerely appreciate your valuable comments and suggestions. Under your guidance, we have made the necessary revisions and improvements to ensure the completeness and eligibility of the paper for publication. We will continue to work diligently to enhance this research and look forward to presenting more valuable research outcomes to you in the future.

Once again, we are grateful for your professional guidance and support.